# Mining Sources of Resistance to Durum Leaf Rust among Tetraploid Wheat Accessions from CIMMYT’s Germplasm Bank

**DOI:** 10.3390/plants12010049

**Published:** 2022-12-22

**Authors:** Julio Huerta-Espino, Ravi P. Singh, Héctor Eduardo Villaseñor-Mir, Karim Ammar

**Affiliations:** 1Campo Experimental Valle de México INIFAP, Apdo. Postal 10, Chapingo 56230, Mexico; 2International Maize and Wheat Improvement Center (CIMMYT), Apdo. Postal 6-641, Mexico City 06600, Mexico

**Keywords:** durum wheat, landraces, *Puccinia triticina*, seedling resistance, adult plant race-specific resistance

## Abstract

A collection of 482 tetraploid wheat accessions from the CIMMYT Germplasm Bank was screened in the greenhouse for resistance to leaf rust disease caused by the fungus *Puccinia triticina* E. The accessions were screened against two races CBG/BP and BBG/BP in the field at two locations: against race CBG/BP at the Norman E. Borlaug Experimental Station (CENEB) located in the Yaqui Valley in the northern state of Sonora in Mexico during the 2014–2015 growing season; and against race BBG/BP at CIMMYT headquarters in El Batan, Texcoco, in the state of Mexico in the summer of 2015. Among the accessions, 79 durum genotypes were identified, of which 68 continued demonstrating their resistance in the field (past the seedling stage) against the two leaf rust races. An additional set of 41 genotypes was susceptible at the seedling stage, but adult plant race-specific resistance was identified in the field. The 79 seedling-resistant genotypes were tested against 15 different leaf rust races at the seedling stage to measure the usefulness of their resistance in a breeding program. Among the 79 accessions tested, 35 were resistant to all races used in the tests. Two sample sources, CIMMYT (18/35) pre-breeding germplasm and Ethiopian landraces (17/35), showed seedling resistance to all races tested except for seven landraces from Ethiopia, which became susceptible to the Cirno race identified in 2017.

## 1. Introduction

Spring durum wheat (*Triticum turgidum* var. *durum*) is grown on about 13 million hectares worldwide, half of which are found in developing countries. Most durum varieties under cultivation in developing countries are either derived from germplasm improvement programs at the International Maize and Wheat and Improvement Center (CIMMYT) or the International Center for Agricultural Research in the Dry Areas (ICARDA). Durum wheat is host to two species of fungi that cause leaf rust: (1) *Puccinia triticina* E., which is the most common and widely distributed [1]; and (2) *P*. *tritici*-*duri* V.-Bourgin [2,3] geographically limited to Morocco [4] and Portugal [5], and most recently in South Spain [6]. Virulence studies conducted from rust pathogen collections made in Chile, Ethiopia, France, India, Israel, Italy, Mexico, Morocco, Spain, Turkey, and the United States [7,8,9,10,11,12,13] determined that *P*. *triticina* races virulent on durum wheat (*T*. *turgidum*) are different from those that are virulent on bread wheat (*T*. *aestivum*). The durum *P*. *triticina* races are avirulent to the majority of the known leaf rust host differentials, carrying resistance genes that are commonly present in bread wheat cultivars; therefore, their survival, selection, and spread are usually restricted to durum growing areas. During the last two decades, new durum leaf rust races with specific virulence have rendered many cultivars susceptible in several countries. In Mexico for example, during the 2001 growing cycle, race BBG/BN [9] defeated the resistance present in the Altar C84 cultivar, which has been grown since 1984. Resistance genes present in Altar C84 have recently been cataloged as *Lr72* [14], which remain highly effective against all leaf rust races, commonly attacking bread wheat in Mexico and elsewhere, including races virulent to *Lr14a* identified in Mexico. The economic losses in Mexico to durum wheat producers because of epidemics from 2001 to 2003 caused by the Altar C84 virulent race BBG/BN were estimated to be over USD 32 million [9]. During 2008, two new durum leaf rust races were identified. Race BBG/BP and CBG/BP both overcome the resistance genes *Lr27*+*31* present in cultivar Jupare and Banamichi, and CBG/BP with additional virulence for *Lr3* defeated the genotype Storlom [15]. In 2017, one of the two races identified in 2008 (BBG/BP) evolved to defeat the resistance gene present in Camayo, which is one of the parents of the cultivar Cirno C2008 [16].

Among the cataloged leaf rust resistance genes originating from durum or other tetraploid *Triticum* species, i.e., *Lr10*, *Lr14a*, *Lr23*, *Lr33* [17], and *Lr72* [14], only *Lr14a* remains effective in Mexico. However, virulence to *Lr14a* among durum leaf rust collections was reported [17] and confirmed by Goyeau et al. [18] in France. Genetic studies, virulence analysis, and molecular mapping have confirmed the presence of resistance genes *Lr14*a [19], *Lr3* [20], *Lr27*+*31* [15], and *Lr16* [21]. These genes are known to be present in bread wheat cultivars. Two additional genes, *Lr61* and *LrCam*, which are unique for durum wheat, were also identified [20,21,22,23] The effectiveness of *Lr3*, *Lr27+31*, and *Lr61* was rapidly lost in Mexico due to the appearance of new virulent leaf rust races. *Lr3* and *Lr61* were not deployed in farmers’ fields, indicating not only the genetic vulnerability of durum wheat to leaf rust in Mexico and other countries, but also the urgent need to search for new sources of resistance genes. 

Durum wheat landraces may take precedence as a good option over wheat wild species in the search for sources of resistance to leaf rust. One example is the resistance found in the Camayo cultivar derived from an Ethiopian landrace and released in Mexico as Cirno C2008 [23]. Cirno C2008 occupied nearly 85% of the durum wheat growing area in Mexico during 2011–2012 and 2012–2013 [24]. Cirno C2008 carries a race-specific major gene located in repulsion with *Lr3* on chromosome 6B [20]. Therefore, a gene designation for this gene is pending. Another example is the identification of two durum wheat landraces (Aus26582 and Aus26579) collected from Portugal that were resistant to leaf rust [25]. From these two accessions, *Lr79* was identified in Aus26582 [26].

It is generally accepted that farmers may have repeatedly selected against extreme susceptibility generation after generation among wheat landraces, and there is no evidence to believe the contrary in the case of durum wheat landrace cultivation. Natural selection could also take place in leaf-rust-prone areas, where the rust affects the vigor and plumpness of host plant seed and hence its rate of reproduction. Therefore, it is likely that during this long, historical pathogen and plant population coexistence, natural selection has favored plant genotypes with an adequate level of resistance and those with race-specific resistance.

In this paper, the resistance of 482 different durum wheat accessions from the CIMMYT Germplasm Bank was evaluated against leaf rust in greenhouse trials at the seedling stage and in field trials at the adult stage at two locations.

## 2. Results

### 2.1. Seedling Tests

Among the 482 accessions tested, at the seedling stage, 69 resistant durum wheat cultivars were identified, from which 68 retained their resistance in the field against the two leaf rust races BBG/BP and CBG/BP [27] at both locations (Table 1). Additional information regarding the accessions such as local name of the landrace or cross in the case of improved material is provided in the Appendix A.

Among the 68 lines, only 35 were resistant to all 12 additional leaf rust races with which they were tested (Table 2). These races included the durum wheat that originated from BBG/BP, CBG/BP, and BBB/BN-61 [27]. 

### 2.2. Adult Plant Tests

Two types of responses were identified in the field tests. One group comprised resistant genotypes whenever hypersensitivity was noticeable, independent of percent infection or final disease severity (Table 1). These were recorded as resistant (R) or moderately resistant (MR). Among the 299 seedling-susceptible lines, 41 showed race-specific adult plant resistance (Table 3), including 14 accessions that showed either winter or facultative growing habit (Table 4).

The second group was classified as susceptible, independent of the final disease severity. A condition for classifying lines into this group was a lack of hypersensitivity and compatible infection type or susceptible response (S). 

Among the durum cultivars evaluated, 258 accessions were seedling susceptible to both races and displayed different degrees of slow rusting type of resistance at the adult stage (data not shown), and 114 seedling resistance were considered as susceptible to both races (data not shown). Therefore, no agronomic value from the disease resistance point of view at this time was considered.

The seedling-susceptible accessions that displayed compatible infection type at the adult stage are not reported in the present study as a different approach to describe them is needed. When grouping the seedling-resistant and seedling-susceptible adult plant-resistant accessions by their origin, they represented 24 countries, the CIMMYT pre-breeding program, and other unknown or non-specified origins. Out of 110 entries, 69 were resistant from seedling to the adult stage (Table 1), and 41 were seedling susceptible, adult plant resistant with hypersensitive response (Table 3 and Table 4).

## 3. Discussion 

Between the two locations used for the adult plant testing, the conditions for rust development at El Batan are more favorable since the growing cycle coincides with the rainy season where more daily moisture is available, and days with prolonged dew formation favor successive reinfections by the fungus. However, in some entries, the level of the disease was higher at the CENEB location since some resistance genes are more effective at lower temperatures, such as *Lr11*, and *Lr18* [28], whereas other genes, such as *Lr13*, could be more effective at higher temperatures [28].

Seedling resistance to all races tested could be the most valuable resistance identified for the short term in a breeding program for countries where both durum and bread wheat are cultivated in the same areas. Several accessions identified as seedling resistant to both durum leaf rust races BBG/BP and CBG/BP were seedling susceptible to at least one race that prefers to attack bread wheat. These genotypes represent a valuable source of resistance against durum leaf rust races, but in this case, resistances identified must be used in combination with other effective sources against all races coming from bread wheat such as *Lr72* present in Altar C84 and Atil C2001 cultivars. 

Recent studies have clearly demonstrated the potential of landraces to contribute diverse alleles for economic traits [32,33,34,35,36,37]. There are several examples of mining sources of resistance among the durum and tetraploid accessions in collections available in germplasm banks [6,33,34,35,36,38,39,40,41,42,43]. Mining these sources of resistance at times are very specific: for leaf rust as an example [6,38,41], yellow rust [39,40,43], stem rust [40,41], or other diseases [37]. However, in only a few cases was resistance screening undertaken with the appropriate leaf rust races [25,38,44]. In fact, in the screening process, the most virulent or the most prevalent race was used to evaluate the resistance of durum genotypes, but usually an isolate or race that preferably attacks bread wheat and not durum [41]. A proposal to rethink the classification of rust races into weak and aggressive categories for hexaploid and tetraploid wheat was suggested [45]. In other words, no leaf rust race specific to durum wheat was used in screening due to the lack of knowledge of the existence of physiological specialization among the leaf rust populations from durum wheat [8]. The use of an inappropriate rust race might lead to classifying resistant genotypes that are indeed susceptible or postulating the wrong resistance gene in other situations [46]. However, if the aim is to search among the durum landraces for leaf rust resistance sources to use in the bread wheat, it is probably correct to test against *P*. *triticina* races coming from bread wheat isolates [47]. The *P*. *triticina* races used in these studies will determine the sources of resistance identified, keeping in mind that in some instances, most durum accessions could be resistant to races collected from bread wheat, but the same accessions could be susceptible to races collected from durum wheat [38]. In other instances, durum wheat and bread wheat are cultivated side by side such as in Mexico, Chile, Argentina, Ethiopia, and North Africa [1], and durum cultivars are required to possess both resistance to races coming from durum wheat and resistance to races coming from bread wheat. Given our objectives to identify sources of resistance to leaf rust to improve the levels and diversity of resistance in durum wheat, it is necessary to test using the race or races that are causing the loss of the resistance in the current durum cultivars. Aoum et al. [38] and Martínez-Moreno et al. [6] have reported similar results. Although most landraces have been reported as susceptible to leaf rust, resistant landraces have been identified that potentially may be used in durum wheat breeding programs [6,34].

Among the seedling-resistant accessions identified in the present study, the majority are true landraces, and therefore, resistance could be considered as unique and different from what has been used in breeding programs already. In other instances, resistance can be traced back either to a bread wheat parent or to a close wild relative, particularly in genotypes from the CIMMYT pre-breeding program. For example, in the seedling resistance lines G5375 and G5369, the presence of RL6010 in the pedigree will allow us to postulate the presence of *Lr9* [28]. The presence of *Lr9* in these accessions was confirmed by inoculating them with a culture virulent to *Lr9* and one avirulent to *Lr9* (data not shown). Entry G5392 is a cross with RL6043/Nacozari, which could indicate the presence of *Lr21* [28]. However, since this gene is in chromosome 1D, the resistance might come from Nacozari M76, a non-identified gene different from *Lr10* [48]. In the line G3543, the resistance might come from the cultivar H567.71, which based on the infection type could be *Lr16*, a leaf rust resistance gene identified in Olesen dwarf or from Santa Elena, both of which are parents of H567.71. Alternatively, *Lr16* may descend from a durum parent since Zhang and Knott [49] postulated its presence in durum cultivars 30 years ago, and more recently, *Lr16* was identified in a *Triticum carthlicum* Blackbird cultivar through molecular markers [21]. The effectiveness of *Lr16* as a race-specific gene, however, is limited since at the adult stage *Lr16* behaves more like a slow-rusting resistance gene—at least against leaf rust races, which preferably attack bread wheat. In bread wheat cultivars, when *Lr16* is alone, disease severity goes from 30MS to 60MS in response to isolates avirulent on seedling (IT 1 to 11^+^). This explains why, among the race-specific resistant seedlings, a few (11/79) were considered as susceptible at the adult stage. Other possible sources of resistance coming from bread wheat are entries G8743 from Sonora 64 (*Lr1*), and G5394 from Nacozari. The resistance in entries G5235, G5207, G5208, G5424, G3988, G6904, G5421, G8813, and G6857 might have come from the wild relatives *T*. *diccocum*, *T*. *boeticum*, *T*. *monococum*, *T*. *carthlicum*, *T*. *rartu*, *T*. *polonicum*, and Khapli-Emmer, respectively, as the durum parents involved in these crosses were susceptible. Among the wild emmer types of wheat, *T*. *dicoccoides* could be considered as the closest wild relative of the durum type [50].

The resistance identified in G8731, G3543, G67857, G6904, G5359, G5364, G5369, and G5394 possibly derives from a bread wheat source effective against all races tested in the seedling stage. However, these sources—although very useful—might be very similar if not the same as the resistance genes *Lr3*, *Lr10*, *Lr11*, *Lr12*, *Lr16*, *Lr23*, *Lr33*, and *Lr27+31*, which are already common in bread wheat germplasm. In the case of G3988, G5424, and G5207, resistance is possibly coming from a wild relative. This resistance is effective against all leaf rust races tested at the seedling stage, which indicates that it might be very useful in the short term for breeding purposes if the aim is to incorporate race-specific resistance genes in breeding durum wheat for leaf rust resistance. 

Among the 17 lines from Ethiopia, which were resistant to all races tested at the seedling stage, except for CWI22214 and CWI20872, 15 lines appeared to belong to *T*. *diccocum* types. These sources of resistance may uniquely belong to the closest wild relative of the *T*. *turgidum* var. *durum* since they are still grown in farmers’ fields in Ethiopia; in fact, the dominance of violet, purple, or red grain color atypical to the more prevalent yellow color of improved germplasm confirms this assumption.

Among the seedling-susceptible adult-resistant plants, two growth habits could be identified: spring and/or winter photoperiod-sensitive types. In both groups, resistance varied from 0 to 40MR, similarly but with less frequency, and resistance coming from a bread wheat source or wild relative was identified. However, most of the resistant genotypes identified were true durum landraces. It is important to notice that, in many instances, lines could not be true landraces, but were introductions from centers of origin or centers of diversification, those whose origin is recorded as being from the Americas when early introductions occurred during the 15th century [51]. 

Independent of the origin of durum wheat genotypes, the study reveals the existence of sources of race-specific resistance to leaf rust in the CIMMYT Germplasm Bank for both seedling resistant-adult resistant and seedling susceptible-adult resistant. In fact, these two sources can be combined to obtain higher levels of resistance and to assure not only diversity, but to increase the number of resistant factors in improved genotypes, which in time will confer higher and stable resistance across environments.

However, the use of these sources of resistance to leaf rust in a durum wheat breeding program will require at least one backcross to an adapted genotype due to the variation in plant height. Most true landraces are typically tall (higher than 100 cm) except for those coming from pre-breeding programs, which are semi-dwarf (less than 100 cm). The second issue will be the red, violet, or purple seed coat color of the grain, and white to mild yellow endosperm. Although simply inherited, the preferred color for improved durum wheat is a very intense yellow pigmentation for not only the seed coat color, but also the endosperm color and its end-use quality of semolina and dry pasta [52].

This relatively small sample of the tetraploid collection from the CIMMYT Germplasm Bank represents a subset of the entire collection. However, the presence of high levels of resistance to wheat leaf rust suggests there is enough variation in the collection, and that in the short term, it is not necessary to look farther into the secondary gene pool for resistance other than into the tetraploid closest relatives such as *T*. *diccocum*, and *T*. *dicoccoides*. The present subset could be used in an association mapping study to understand the degree of diversity among the resistant accessions against the durum leaf rust races in different countries and environments. Resistant landraces and pre-breeding genotypes identified in the present study will complement the efforts of searching for new sources of resistance and their use in breeding durum programs against leaf rust.

The response of the resistant lines to the new race virulent on Cirno C2008 [16] indicated that a few genotypes might carry the same resistance factor(s) as Cirno C2008 (*Lr Camayo*). This is evidenced by the following previously resistant lines becoming susceptible to the Cirno race: ABYSSINIAN 26 (CWI23065), ELS6404.131.3 (CWI23440), ELS6404.61.2 (CWI23473), HARLAN J.R 1939 (CWI21737), IAR.W.63.1 (CWI22175), IAR.W.84 (CWI22201), and IAR.W.92.1 (CWI22214). All these landraces were collected from Ethiopia. The defeated gene present in the Camayo parent of Cirno C2008 has an Ethiopian landrace origin—ETH-LRBR A1-133/3*ALTAR 84 [23] also. When the Altar gene (*Lr72*) was defeated in 2001 [9], Camayo was resistant. All other evaluated landraces from different origins remained resistant to the new leaf rust race. 

## 4. Materials and Methods

### 4.1. Seedling Tests

Seedling resistance was determined by evaluating 482 durum wheat accessions from CIMMYT’s germplasm bank against two of the most recently identified leaf rust races, which have evolved in Mexico since 2001. These accessions comprised landraces and pre-breeding lines from 45 countries and included pre-breeding accessions coming from crosses made at CIMMYT. 

For the greenhouse evaluation, 8 to 10 seeds of each line were sown in plastic trays filled with steam-sterilized soil. Plants were grown in a greenhouse at 18 to 22 °C with 16 h of supplemental light. Fourteen-day-old seedlings at the two-leaf stage were inoculated by spraying fresh urediniospores of *P*. *triticina* races BBG/BP and CBG/BP [27] suspended in non-phytotoxic mineral oil Soltrol 170^®^ (Phillips Petroleum Company, Borger, TX, USA) at a concentration of 5 mg/mL [20]. After air drying for 30 min, the inoculated seedling plants were incubated in a mist chamber for 16 to 20 h at 22 °C and 100% relative humidity and then moved to the greenhouse for further incubation. Infection-type responses were recorded about 14 days after inoculation using the 0–4 scale described by Roelfs et al. [28]. In this scale, infection types 3 and 4 are categorized as susceptible responses, and the remaining infection types are considered as resistant.

The differences between CBG/BP and BBG/BP are found only in *Lr3* based on the North American system of nomenclature [29] and additional sets describing the variation in Mexico [30]. Their identical avirulence/virulence formula for other genes are *Lr1*, *2a*, *2b*, *2c*, *3bg*, *3ka*, *9*, *13*, *14a*, *15*, *16*, *17a*, *18*, *19*, *21*, *24*, *25*, *26*, *28*, *29*, *30*, *32*/*Lr10*, *12*, *14b*, *17b*, *20*, *23*, *27+31*, and *Lr33* [27].

Seedling-resistant genotypes against BBG/BP or CBG/BP races were further evaluated against an additional race from durum wheat: BBB/BN-*Lr61*, which is avirulent to cultivar Altar C84 (*Lr72*), but virulent to *Lr28*, Gaza, and *Lr61*. Besides the durum leaf rust races CBG/BP, BBG/BP, BBB/BN-*Lr61*, an additional 12 different rust races isolated from bread wheat—CBJ/QB, CBJ/QL, CBJ/QQ, MCJ/SP, MCJ/QM, MCD/SN, MBJ/SP, NCJ/BN, MFB/SP, MBB/QN, MBB/BN, and TCB/TD—were included in the test, following the same procedure described above and used in the seedling test against races BBG/BP and CBG/BP.

### 4.2. Adult Plant Tests

To accurately measure the response to the two most common durum leaf rust races—CBG/BP and BBG/BP—whose avirulence/virulence formula is described above, the 482 durum wheat lines were planted at two locations and inoculated with different races at each location. The 482 accessions were sown in the field in plots of paired rows 1 m long from 20 to 25 November 2014 at the Norman E. Borlaug Experimental Station (CENEB) located in the Yaqui Valley in Sonora, Mexico. The experiment was sown on raised beds spaced at 80 cm intervals. Spreader rows of the susceptible variety ‘Storlom’ wheat (*Lr3* carrier) were planted on each side of the experimental plots and as hills in the alleys between plots to assure the leaf rust artificial epidemic inoculum was distributed evenly. The plots were protected from weeds and insect pests with agrochemicals applied as necessary. They were well managed with N, P, and K fertilization, and irrigated until all lines reached physiological maturity. Spreader rows and hill plots were inoculated with urediniospores of the leaf rust fungus race CBG/BP on 20 January 2015. Inoculation was repeated thrice to assure the onset of the epidemic. Adult plant leaf rust severity and response to infection were recorded when the susceptible check plots of Atil C2001, planted every 100 rows, displayed approximately 80 to 100% leaf rust severity. The rust severity was based on the modified Cobb Scale [31]. Disease development on the lines was first recorded on 10 March, then on 22 March, and finally on 5 April 2015. 

A second field test was planted at the CIMMYT headquarters El Batan research station near Mexico City from 15 to 25 May. The spreader rows were a mixture of the durum cultivars Jupare C2001 and Banamichi C2004, which both carry *Lr27+31* resistance genes [15]. They were planted on both sides of the experimental plots and as hill plots in the alleys to assure the leaf rust artificial epidemic inoculum was spread evenly. The plots were protected from weeds and pests using agrochemical applications as necessary and were well managed with N, P, and K fertilization. Irrigation was only required for germination since the growing season at El Batan coincides with the rainy season. Spreaders were inoculated on 28 and 29 June by spraying fresh urediniospores of the leaf rust fungus race BBG/BP suspended in the lightweight mineral oil (Soltrol 170^®^). Adult plant leaf rust severity and response to infection were recorded when the susceptible check plots of Atil C2001 planted every 100 rows displayed approximately 80 to 100% leaf rust severity. The severity estimation was based on the modified Cobb Scale [31]. 

In both field experiments, plant height, days to heading, grain color, growth habit, and the presence of leaf tip necrosis (LTN), were recorded. 

## 5. Conclusions

In the present study, high levels of diversity for leaf rust and other agronomic attributes were found among the core subset of durum landraces and pre-breeding lines from the CIMMYT durum wheat collection. High levels of resistance to durum wheat leaf rust identified at the seedling and adult stages suggest that there is enough variation in the collection, and that in the short term, it is not necessary to look farther into the secondary gene pool for resistance, other than into the tetraploid closest relatives such as *T*. *diccocum*, and *T*. *dicoccoides*. The sources of resistance to leaf rust here identified at the seedling and adult stages can be combined to obtain higher levels of resistance and to assure not only the diversity, but to increase the number of resistant factors in improved genotypes, which in time will confer higher and stable resistance across environments. Several resistant sources identified in the present study are already incorporated in durum lines by the durum wheat breeding program at CIMMYT, which in the short term will enhance the gene pool diversity against *P*. *triticina* E. the causal agent of the wheat leaf rust.

Small seed samples of all accessions are available upon request to the first author or through CIMMYT (seed request by Intrid, CIMMYT Germplasm Bank accession number).

## Figures and Tables

**Table 1 plants-12-00049-t001:** Infection types at seedling (GH = greenhouse), final disease severity in the field in response to two leaf rust races, and plant height of selected seedling resistant.

		Response to Leaf Rust Races #				Plant	
	Country	CBG/BP	CBG/BP	BBG/BP	BBG/BP	Height	Remarks
Intrid ^a^	Origin (Seed)	GH ^b^	Field ^c^	GH	Field	cm	
G5375	CIMMYT	0;	0	0	0	85	
G5369	CIMMYT	0;	0	0	0	90	
CWI20891	SPAIN	0;	0	X	0	130	1
CWI22027	CYPRUS	0;	0	0;	0	150	
DW5345	TURKEY	0;	1R	0;	5R	145	1
G5359	CIMMYT	1	0	1	0	70	
G5392	CIMMYT	;12	0	12	0	75	
DW15671	CIMMYT	;1−	0	;1−	0	85	
G5364	CIMMYT	1	0	1	0	90	
DW15675	CIMMYT	;1=	0	X−	0	90	
CWI21151	PORTUGAL	;1	0	X	0	105	
CWI20528	TURKEY	1	0	1	0	115	
CWI23230	ETHIOPIA	;1	0	;1	0	115	
CWI22327	JORDAN	;1	0	;1	0	130	
CWI32536	GREECE	;1	0	;1	0	140	
G6857	CIMMYT	12	0	12	20R	55	
G3543	CIMMYT	;1	5MS	;1	0	50	
G8743	CIMMYT	11+	5MR	1	10MR	65	
CWI22214	ETHIOPIA	;1	5R	1−	0	105	
CWI23460	ETHIOPIA	;1−	1MR	;1−	5MR	115	1, 2
DW7085	CIMMYT	12	5R	12	20R	125	
G8731	CIMMYT	1	5MS	1	40S	65	
G5235	CIMMYT	X+	0	X+	0	70	2
G5207	CIMMYT	X+	0	X+	0	75	
G5394	CIMMYT	X	0	;	0	80	
DW15673	CIMMYT	X−	0	X−	0	85	
G8505	CIMMYT	X	0	X + 3	0	105	1
CWI22423	SOUTH AFRICA	X=	0	X=	0	115	
CWI21666	GREECE	X=	0	X	0	120	
CWI20476	TURKEY	X	0	X	0	130	1, 2
CWI22683	RUSSIA	X-	0	X=	0	140	1
CWI22087	ETHIOPIA	X	-	X	-	-	1
CWI21791	RUSSIA	X-	0	X−	5MR	155	
CWI22053	ISRAEL	X-	0	;1	15MR	120	
G5208	CIMMYT	X	0	X	20R	70	
G5424	CIMMYT	X−	5R	X	0	95	2
DW649	CIMMYT	X=	15MR	X	0	80	
G3988	CIMMYT	X	10R	X	0	90	
CWI355	MEXICO	X	10MR	X	10MR	155	1
G5421	CIMMYT	X	20MR	X+	20MR	55	
CWI23606	USA	X	5MR	X	20MR	110	3
CWI21696	YUGOSLAVIA	X + 3	0	X + 3	0	100	2, 4
G8813	CIMMYT	2	5R	2	5R	60	
DW7103	CIMMYT	2	15MR	2	15MR	130	
CWI22175	ETHIOPIA	;1−	0	;1−	0	125	2
CWI20114	TURKEY	;1−	0	;1−	0	130	2
CWI23067	ETHIOPIA	;1	0	;1	0	135	2
CWI20425	TURKEY	;1	0	;1	0	135	2
CWI21256	ETHIOPIA	X	0	X	10R	110	2
CWI22139	ETHIOPIA	12	0	12	20R	135	2
CWI22166	ETHIOPIA	X-	0	X−	0	105	1, 3
CWI22201	ETHIOPIA	X+	0	X	0	105	3
CWI22089	ETHIOPIA	X	0	X	0	110	3
CWI22102	ETHIOPIA	X+	0	X+	0	115	3
CWI22280	ETHIOPIA	X+	0	X	0	115	3
CWI22064	ETHIOPIA	X	0	X	0	120	3
CWI22294	ETHIOPIA	;1	0	;1	0	120	3
CWI22250	ETHIOPIA	11+	0	11+	30R	110	3
CWI23359	SPAIN	22+	0	12	0	140	
DW3139	TURKEY	;1−	0	;1	0	110	4
CWI23446	ETHIOPIA	X+	1R	X+	5R	115	3
G6904	CIMMYT	1	5R	1	0	60	2
CWI21737	ETHIOPIA	;1	5R	;1	0	105	2
CWI23440	ETHIOPIA	;1−	5R	;1=	5R	120	2
CWI22143	ETHIOPIA	X	5MR	X	0	110	3
CWI23065	ETHIOPIA	;1	5R	;1	0	115	3
CWI23385	ETHIOPIA	X	5MR	X	0	120	3
CWI23473	ETHIOPIA	X + 3	15MR	X + 3	10MR	115	2

Intrid ^a^ = CIMMYT Germplasm Bank accession number. ^b^ = Greenhouse infection type follows Roelfs et al. [28] ^#^ Race nomenclature follows Long and Kolmer [29] and Singh [30] ^c^ = Leaf rust severity follows Peterson et al. [31]. Remarks: 1 = Leaf tip necrosis (LTN); 2 = Red grain; 3 = Purple or violet grain; 4 = Winter or photoperiod sensitive.

**Table 2 plants-12-00049-t002:** Infection types of tetraploid accessions in response to 15 leaf rust races at seedling stage in the greenhouse.

		Leaf Rust Races ^#^														
	Country	CBG/	BBG/	BBB/	CBJ/	CBJ/	CBJ/	MCJ/	MCJ/	MCD/	MBJ/	NCJ/	MFB/	MBB/	MBB/	TCB/
Intrid	Origin	BP ^d^	BP ^d^	BN ^d^	QB	QL	QQ	SP	QM	SN	SP	BN	SP	QN	BN	TD
DW7085	CIMMYT	12	12	X	;	;	;	;	;	;1=	;	;1=	;	;	;	;1
DW7103	CIMMYT	2	2	X	;	;	;	;	;	;	;	;	;	;	X	;
DW15671	CIMMYT	;1−	;1−	;	0;	0;	;	;	0;	0;	;	;	;	;	;	;
DW15673	CIMMYT	X−	X−	0;	0;	0;	;	;	0;	0;	0;	;	;	;	;	0;
DW15675	CIMMYT	;1=	X−	0;	0;	0;	;	;	0;	;	;	;	;	;	;	;
G3988	CIMMYT	X	X	XCN	;C	;C	;C	;C	;CN	;CN	;1 − C	;	;	;1 = C	;1 = C	;1 − C
G5424	CIMMYT	X-	X	XCN	11+	;	11+	;	X=	11+	11+	11+	11+	11+	X	11+
G8731	CIMMYT	1	1	X	; ^e^	1	X−	;	;	;	;	X=	;	;	;	;
G3543	CIMMYT	;1	;1	1	;1−	11+	;	;1	;1−	;1=	;1−	;1=	;1−	;1−	;1=	;1
G6857	CIMMYT	12	12	1	;1−	;1	;1=	;	;	;	;	;1=	;	;	X−	;1
G6904	CIMMYT	1	1	X	;1	;1	;1	;1	;	;1	;1	;1	X	X	;	;1
DW7147	CIMMYT	1	1	X	;	;1	X	;1−	;	12	X−	;1	;1	X	11+	;1
G5359	CIMMYT	1	1	1	;1−	1	;1	1	;1−	;1=	1	X=	;1−	X=	1	;1
G5364	CIMMYT	1	1	X=	1	1	0;1	1	;1=	;1=	1	X=	;1−	X=	;1	;
G5207	CIMMYT	X+	X+	XCN	X	X	1CN	X	;1C	X	1	X	X	X=	XCN	X
G5369	CIMMYT	0;	0	0	0	0	0;	0;	0	;	;	0	0	0	0	0;
G5375	CIMMYT	0;	0	0	0	0	0	0	0	0	0	0	0	0	0	0
G5394	CIMMYT	X	;	X	11+	11+	X	X	11+	1	11+	X	11+	11+	X	1
CWI23065	ETHIOPIA	;1	;1	;	;	;	;	;	;	;	;	;1=	;	;	;	;
CWI23440	ETHIOPIA	;1−	;1=	XCN	;	;	;	;	;	;1=	;1	;1	;1=	;1=	X	;1
CWI23446	ETHIOPIA	X+	X+	X	;C	X = CN	;C	X=	X = CN	X = CN	X	X	X	X	X	X
CWI23473	ETHIOPIA	X + 3	X + 3	;	X	X	X	X	X=	;	;1	X	;1	X=	X	;1
CWI23385	ETHIOPIA	X	X	X+	X	X	X	;	X−	X	X	X	X	X	X	;
CWI21256	ETHIOPIA	X	X	X-	;1=	;	11+	;C	;	;1	;C	X=	X=	;1=	;1=	;C
CWI21737	ETHIOPIA	;1	;1	X	;	;1	;	;	;	;	;	;	;	;	X=	;
CWI22064	ETHIOPIA	X	X	X	X	X	X	X	;C	X	X	X	XCN	XCN	X	X
CWI22089	ETHIOPIA	X	X	X	X	X	X	;	X	X	X	X	X	X	X	X
CWI22294	ETHIOPIA	;1	;1	X	X	;	;1=	;	;	X	;	;1=	;	;	;	;
CWI22102	ETHIOPIA	X+	X+	X	X	X	X	;	X	X	X	X	X	X	X	;1
CWI22143	ETHIOPIA	X	X	X	;	;	11+	X	;	X	11+	11+	X	X	X	11+
CWI22166	ETHIOPIA	X−	X−	X	X	X=	X−	X	;	X	X	X	X	X	X	X
CWI22175	ETHIOPIA	;1−	;1−	;1	;	;	;1=	;	;	;	;	;1−	;	;	X	;
CWI22201	ETHIOPIA	X+	X	X	;1	;	;	;	;	;	;	;1−	;	;1=	X=	;1
CWI22214	ETHIOPIA	;1	1−	X−	;	;	;	;	;	;	;	;	;	;1=	;1=	;
CWI20872	ETHIOPIA	;1	;1	;	;	0;	;	;	;	;	;	;1	;	;	;	;

Intrid = CIMMYT Germplasm Bank accession number. ^#^ Race nomenclature follows Singh [30]. ^d^ BBG/BP, CBG/BP, and BBB/BN are typically durum leaf rust races [27]. ^e^ Infection type follows Roelfs et al. [28] (X = mesothetic response; C = chlorosis).

**Table 3 plants-12-00049-t003:** Infection types at seedling, final disease severity in the field in response to two leaf rust races, and plant height of selected seedling susceptible adult plant resistant tetraploid accessions.

		Leaf Rust Races ^#^					
Intrid ^a^	Country	CBG/BP	CBG/BP	BBG/BP	BBG/BP	Height	
	Origin (Seed)	GH ^b^	Field ^c^	GH	Field	cm	Remarks
G5202	CIMMYT	3+	0	3+	0	65	
DW5199	ETHIOPIA	4	0	3+	0	125	3
CWI20826	SPAIN	3+	0	3+	0	125	3
DW4477	SPAIN	4	0	3+	0	155	1, 3
CWI13539	INDIA	3+	0	3+	0	150	3
CWI21033	PORTUGAL	4	0	3+	0	160	3
CWI19754	ITALY	4	0	3+	0	140	3
CWI19758	PORTUGAL	3	0	3+	0	115	2, 3
CWI26558	BRAZIL	3+	0	3+	0	130	2, 3
CWI22230	ETHIOPIA	4	0	3+	0	125	1, 3
CWI22921	USSR	4	0	3+	0	155	3
CWI22927	USSR	4	0	3+	0	155	3
CWI22466	UAR	4	0	4	5R	115	3
CWI23095	MOROCCO	4	0	3+	20R	150	2
CWI23602	USA	4	0	4	5R	135	2, 3
CWI75601	ETHIOPIA	4	0	3+	15R	115	4
DW4192	PORTUGAL	4	1MS	3+	5R	130	2, 3
G7358	CIMMYT	3+	5R	3+	30R	80	3
G7492	CIMMYT	3+	5MR	3+	30R	55	
CWI21086	ETHIOPIA	3+	20R	3+	30MR	125	1, 3
CWI20285	MEXICO	4	20MR	4	5R	120	
CWI23075	UAR	4	0;	3+	20MR	115	1, 3
CWI22396	FRANCE	4	0	3+	40MR	75	1
CWI20657	INDIA	4	20MR	3+	40R	125	
CWI22951	UAR	4	30MR	4	30MR	120	

^a^ = Intrid = plant introduction ID or CIMMYT Germplasm Bank accession number. ^#^ = Race nomenclature follows Singh [30]. ^b^ = Infection type follows Roelfs et al. [28]. ^c^ = Leaf rust severity follows Peterson et al. [31]. Remarks: 1 = Dicoccum type; 2 = Leaf tip necrosis; 3 = Red grain; 4 = Purple or violet grain.

**Table 4 plants-12-00049-t004:** Infection types ^a^ at seedling, final disease severity in the field in response to two leaf rust races, and plant height of selected seedling-susceptible adult plant-resistant winter or facultative tetraploid accessions.

		Leaf Rust Races ^b^					
Intrid ^#^	Country	CBG/BP	CBG/BP	BBG/BP	BBG/BP	Height	
	Origin (Seed)	GH ^c^	Field ^d^	GH	Field	cm	Remarks
DW5442	TURKEY	4	0	3+	0	120	
CWI22788	SPAIN	4	0	3+	0	145	
DW4212	UK	4	0	3+	0	65	3
CWI334	MEXICO	4	0	3+	0	110	3
CWI22799	MEXICO	4	0	3+	0	155	2
CWI356	MEXICO	4	0	3+	0	125	3
CWI20224	TURKEY	4	0	3+	0	125	2
CWI22966	RUSSIA	4	0	3+	0	140	1, 3
CWI22789	SPAIN	4	0	3+	5R	145	
CWI22035	RUMANIA	4	0	3+	5R	145	
CWI23196	USSR	4	0	3+	5R	145	
CWI22844	PORTUGAL	4	5R	3+	5R	145	
DW4219	UK	4	5R	3+	5R	100	3
CWI20330	EGYPT	4	40MR	3+	40MR	115	3

^#^ = Intrid (CIMMYT Germplasm Bank accession number). ^a^ = Infection type follows Roelfs et al. [28]. ^b^ = Race nomenclature follows Singh [30]. ^c^ = Leaf rust severity follows Peterson et al. [31]. Remarks: (additional information ^d^) 1 = Dicoccum type; 2 = Leaf tip necrosis; 3 = Red grain.

## Data Availability

Data is available upon request to the first author. Data remains in the field books with different TID’s.

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
