# Peer review of "Mining Sources of Resistance to Durum Leaf Rust among Tetraploid Wheat Accessions from CIMMYT’s Germplasm Bank"

_plants, 2022, doi:10.3390/plants12010049_

Round 1

Reviewer 1 Report

Overall this manuscript is a good addition to the literature. It presents relevant information on resistance to durum leaf rust that can be used to develop resistance durum wheat cultivars. There are some sections that could be clarified so that the reader can better understand the procedures that the authors used. Specifically the abstract is confusing as to what tests were done and what the results were from those tests.

Lines 11-14 “ The accessions were screened against two races CBG/BP and BBG/BP in the field at two locations: 11 Against race CBG/BP at the Norman E. Borlaug Experimental Station (CENEB) located in the Yaqui 12 Valley in the northern state of Sonora in Mexico during the 2014-2015 growing season; and against 13 race BBG/BP at CIMMYT headquarters in El Batan, Texcoco, in the state of Mexico in the summer 14 of 2015.

Should be changed to;

These accessions were screened against two races in the field at two locations: against race CBG/BP at the Norman E. Borlaug Experimental Station (CENEB) located in the Yaqui Valley in the northern state of Sonora in Mexico during the 2014-2015 growing season; and against race BBG/BP at CIMMYT headquarters in El Batan, Texcoco, in the state of Mexico in the summer of 2015.

Lines 15-16 “Among the accessions, 79 durum genotypes were identified with seedling resistance of which 68 continued demonstrating their resistance in the field (past the seedling stage) against the two leaf rust races.

Lines 18-23 “ The 79 seedling resistant genotypes were tested against 15 additional bread wheat leaf rust races at the seedling stage in order to measure the usefulness of their resistance in a breeding program. Among the 79 accessions tested, 35 were resistant to all races used in the tests, of which 18 were CIMMYT pre-breeding germplasm lines and 17 were Ethiopian landraces. These had seedling resistance to all races tested except for seven landraces from Ethiopia which were susceptible to the Cirno race identified in 2017.

The other races used were identified using the five letter code, eg. BBG/BP but here the race was just referred to as the Cirno race. The authors should add the five letter code for the Cirno race to avoid confusion.

Lines 65-70 The authors jump back and forth between durum and bread wheat with the resistance genes. It is important to state in each sentence whether they are referring to durum or bread wheat to avoid confusion.

The authors describe the Cirno race in both the abstract and the discussion but it is not mentioned in the methods. Was this race tested in this study? If so the methods should be described. If this was from a different study the text and the abstract need to state that.

Line 163 The authors state here that 69 durum wheat cultivars were resistant at the seedling stage but in the abstract they state 79 resistant genotypes. Which is correct?

Lines 291-221 The authors discuss the two locations for adult plant testing however the figures list the race used at each location rather than the locations. The authors should add which race was used at each location to their discussion at this point for clarification.

Lines 328-333 The authors discuss the reaction of these lines to the new race virulent on Cirno C2008. This appears to have been done in a previous study so the results should not appear in the abstract of this study. The authors should also state the five letter code of this race and explain how it differs from the two races used in this study.

Author Response

Please see  an attached file

Reviewer 2 Report

Manuscript describes the screening of 482 tetraploid durum wheat genotypes against two leaf rust pathotypes at the seedling and adult plant stage. 79 accessions were identified to be resistant at the seedling and adult plant stage while 41 accessions were found to possess adult plant resistance genes. 35 out of 79 R  accessions were resistant against 15 other leaf rust pathotypes. Overall, manuscript is very well written except few grammatical errors. 

I have few queries 

1. Two leaf rust races CBG/BP and BBG/BP were used for seedling screening under greenhouse conditions. Authors have stated that adult plant screening for same races was done under field conditions. Under open field conditions other leaf rust races might also be present even if inoculations were done with CBG/BP and BBG/BP. Kindly clarify

2. Line 120-121, 114 seedling resistant accessions were susceptible at the adult plant stage. Seedling resistant accessions are expected to maintain their resistance throughout their life. The seedling data should be checked again carefully to rule out any escapes. 

3. Why plant height is being included in all tables. How is it linked to the resistance/susceptibility

Author Response

please see an attached file

Reviewer 3 Report

Tables are difficult to understand. Title can be modified.

Author Response

please see an attached file

Reviewer 4 Report

Line 260; “However, since this gene is in chromosome 1D,” Replace in with on.

Table 2; change the formatting to make it easier to read. Potentially changing the orientation to landscape will be sufficient.

What the numbers represent in the “Remarks” columns in tables 1, 3 and 4 changes from table to table. Keep this consistent across tables, so that 1 always equals LTN, 2 = Red grain and so on.   

Author Response

please see an attached file
